# A STUDY OF BLACK-BOX ATTACKS AGAINST ROBUST FEDERATED LEARNING

## ABSTRACT

The original Federated Learning (FL) algorithm, FedAvg, is vulnerable to adversarial attacks from its clients. To enhance the security of FL, researchers introduced various defensive aggregation rules. Some of the aggregation rules are based on robust statistics, such as geometry median, and Krum, and some are designed against Sybil attackers, namely FoolsGold, and CONTRA. The previous works evaluate their robustness in a white-box setting, where attackers know which aggregation rule is used by the federated server, the parameters of the FL system, and sometimes the data of honest clients. In this paper, we propose an untargeted attack algorithm based on reinforcement learning (UA-RL) to study the robustness of the aggregation rules in a black-box setting. UA-RL uses the sum of gradients of unmodified datasets to maximize the loss function. It applies reinforcement learning to search for the best parameter controlling the attack magnitude to bypass the aggregation rules. Our experiments on non-i.i.d. datasets indicate that defensive aggregation rules, including Krum, geometry median, FoolsGold, and CONTRA are vulnerable to UA-RL attacks. On i.i.d. datasets, FoolsGold, and CONTRA are fragile, but geometry median and Krum are relatively robust. We further perform a theoretical analysis to explain these experiment results.

## 1 INTRODUCTION

The rapid development of Federated Learning raises concerns about its security and reliability in a real-world environment. Researchers have studied two types of adversarial attacks against FL: untargeted and targeted attacks. The untargeted attack aims to damage the performance of the global model, while the targeted attack tries to mislead the global model on a specific task that benefits malicious groups.

To alleviate the impact of these attacks, researchers have developed various countermeasures and robust methodologies. Zhang et al. (2022) used secure aggregation to detect backdoor attackers. Ozdayi et al. (2021) implemented an adjustable learning rate on the federated server to defend against backdoor attacks. Meng et al. Meng et al. (2021) created a visualization of the FL operation, which uses human analysts to spot unusual clients from a user interface. Besides these approaches, the majority of the defense methods are statistical-based aggregation rules. The original averaging aggregation from the FedAvg McMahan et al. (2017) is replaced with an aggregation rule that has a defensive feature. These defensive aggregation rules use different schemes to estimate the reliability of the information shared by each client. Then a weighted sum is performed on the server side to determine clients' weights according to the reliability.

Compared with other methods, the statistical-based aggregation rules are easier to implement and more cost-effective. They require no extra training data for the server nor knowledge about the learning task. Their computation overhead is usually small. Krum Blanchard et al. (2017), Bulyan El Mhamdi et al. (2018), coordinate-wise median Yin et al. (2018) and geometric median Chen et al. (2017) are median-based rules that remove or reduce the effects from clients who are statistical outliers. FoolsGold Fung et al. (2020) and CONTRA Awan et al. (2021) were designed based on the behavior of Sybil attackers. They remove malicious by identifying clients with high similarity. Other defenses like CRFL Xie et al. (2021), FLCert Cao et al. (2022), and Auror Shen et al. (2016),

exploited multiple statistical tools, such as clipping, smoothing, and clustering to defend targeted poisoning attack and backdoor attack.

Previous works Li et al. (2022) Baruch et al. (2019) Xie et al. (2020) Fang et al. (2020) rely heavily on prior knowledge of the FL system to break some of these robust aggregation rules in a white-box setting. However, this prior knowledge may not be available in a real-world environment. In addition, a thorough analysis of the vulnerability of some of these aggregation rules has not been conducted. In this paper, we propose an untargeted attack algorithm based on reinforcement learning (UA-RL) to study FL aggregation rules' robustness in a black-box setting. UA-RL uses the sum of gradients of unmodified datasets to maximize the loss function. It applies reinforcement learning (RL) to search for the best parameter controlling the attack magnitude to bypass the aggregation rules. In addition, we conduct theoretical analyses on Krum and FoolsGold. The contributions of our work include:

- To our best knowledge, we are the first to develop a black-box attack against FL. UA-RL, our model-free reinforcement learning based attack is effective on the robust aggregation rules, including Krum, geometric median, FoolsGold, and CONTRA.
- We experimentally indicate the limitations of the existing robust aggregation rules, in particular on non-i.i.d. datasets, and show that UA-RL adapts its attack magnitudes to attack different aggregation rules.
- We conduct a theoretical study to explain the experimental findings and analyze the vulnerability of the aggregation rules.

## 2 Preliminary and Related Works

### 2.1 Federated Learning Algorithm

FL trains a global model on distributed databases. Firstly, a FL server initializes a global model and sends it to its clients. The clients train the model for some iterations on their local datasets and then send the model weights back to the server. The server takes the weights from the clients and uses an aggregation rule and the server's learning rate to update the global model. Then server sends the updated global model back to the clients again. This process repeats until the global model reaches convergence. In this paper, $w^t$, $w_i^t$, $Aggr$ and $\bar{w}^t$ represent the global model weights, the model weights from the $i^{th}$ client, an aggregation rule, and aggregated weights at the $t^{th}$ iteration, respectively. More clearly, $\bar{w}^t = Aggr\left(w_1^t \cdots w_n^t\right)$ and the server uses $w^{t+1} = w^t - lr_{sever}(w^t - \bar{w}^t)$, where $lr_{sever}$ is the learning rate of server for updating the global model. The local model's update at the $t^{th}$ iteration is denoted as $Q_i^t = w_i^t - w^t$, which is called the sum of the negative gradient because the negative gradient is used in each local SGD iteration. Algorithm 2 in the appendix summarizes the FL training process and Table 4 in the Appendix lists the symbols used in this paper.

### 2.2 Statistical-based Aggregation Rules

Based on their design principles, statistical-based aggregation rules can be categorized into two groups: central tendency and outer tendency. Krum, coordinate-wise median, and geometric median rely on robust statistics that have a central tendency. FoolsGold and CONTRA are designed to defend against Sybil attacks, where the clients controlled by a malicious group have very similar behaviors, compared with honest clients. To reduce the impact of these Sybil clients, FoolsGold and CONTRA have an outer tendency for over similar clients. These two groups of aggregation rules are based on contradictive design principles. One favours highly similar clients and the other reduces impacts from highly similar clients. Although they both use common similarities and dissimilarity measures, such as L2 distance and cosine similarity to measure the difference between clients, because of their contradictive design principles, they have very different characteristics.

### 2.3 Attack the Federated Learning

Research on how to attack Federated Learning's defense methods helps us understand the characteristics of the defense and contributes to creating better defense strategies. Blanchard et al. (2017)

proved that with a linear aggregation rule such as FedAvg, a single Byzantine worker can prevent convergence. After Byzantine-robust FL methods had been introduced, effective attack strategies were found in a full-knowledge, white-box setting. Fang et al. (2020)'s experiments on poisoning attacks showed that Trimmed Mean Welsh (1987), Krum, and coordinate-wise median could be broken using local model poisoning when the aggregation rule is known to the malicious party. In a grey box setting, the malicious party knows the aggregation rule but can not access the models of the honest clients. Fang et al. used an estimation based on Gaussian distribution and binary search for the attack parameters. Baruch et al. (2019) found through experiments if the variance between honest clients is high enough, a moderate attack within the population variance would break Trimmed Mean, Krum, and Bulyan. As a white-box setup, Baruch et al. required an analysis of the gradients from honest clients to configure the attack parameters. Li et al. (2022) created an RL-based FL attack framework in a grey-box setting, where the FL system's parameters and aggregation rule were known to the malicious, but datasets of honest clients were unknown. The malicious group utilizes the given knowledge to recreate a separate FL process as a simulator of the FL environment. With a model-based RL agent, the malicious group learns the data distribution and policy for attacking and then carries out the attack based on the learned policy. These previous methods assume that the malicious group knows which aggregation rule the FL server has in place. Based on the properties of the aggregation rules, the malicious party could implement a targeted attack strategy. In contrast, our proposed method UA-RL operates in a black-box setting, where the aggregation rule is unknown to the malicious group.

## 3 METHODOLOGY

### 3.1 PROBLEM SETUP

In this study, we focus on a scenario that is closer to a real-world FL operation. Our assumptions on the FL system and the threat model are summarized below:

- The malicious group does not know the parameters of the FL system, including the total number of clients, the aggregation rule, and the total iterations of FL, nor does it have access to data from other honest clients.

- The goal of the attack is to decrease the global model's accuracy and prevent convergence.

- Malicious clients work in collusion and operate as one group. The group owns the local data of all malicious members.

- The server does not own any training data.

- The server's learning rate ($lr_{server}$) is a reasonably small number and stays unchanged during the iterations.

### 3.2 THE PROPOSED UA-RL

Once a malicious client $a$ receives the server model weights, $w^t$, they train it on an unmodified local dataset and obtain $w_a^t$, which is supposed to be sent back to the server. The malicious client first computes the difference between the updated model and the server model, i.e., $\Psi_a^t = w_a^t - w^t$. We call $\Psi_a^t$ as the sum of negative gradients because if SGD is used, $\Psi_a^t$ is equal to $-lr_{local} \times \sum_{i=0}^{J} \nabla F(w_a^j)$, where $w_a^0 = w^t$, $F$ is a loss function, and $lr_{local}$ is a constant local learning rate. The training rule for SGD is $w_a^{j+1} = w_a^j - lr_{local} \times \nabla F(w_a^j)$. Note that $w_a^t = w_a^{J+1}$, is the one obtained from the local training. It should be emphasized that the malicious clients do not necessarily need to know the loss function, local learning rate, and the local training process. They only need to access the locally updated weights $w_a^t$ and the server weights, $w^t$. Once $\Psi_a^t$ is computed, the malicious client modifies the updated weights as $\widetilde{w}_a^t = w^t - \beta \Psi_a^t$, where $\beta$ called attack magnitude is determined by an RL algorithm. $\widetilde{w}_a^t$ is sent back to the server. UA-RL applies the same attack strategy and same attack magnitude, $\beta$ to all malicious clients, but their attack updates are trained on different local datasets. It can be observed that the modified update aims to maximize the loss function. More clearly, $\widetilde{w}_a^t = w^t - \beta \Psi_a^t = w^t + \beta lr_{local} \times \sum_{i=0}^{J} \nabla F(w_a^j)$, which can be considered as a gradient ascent algorithm to maximize the loss function. The principle of this attack scheme is shown in Figure 1.

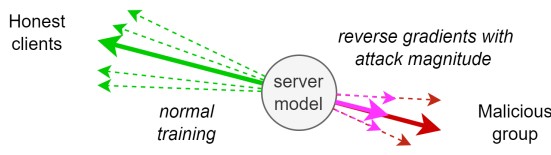

Figure 1: Reversing gradient attack with RL adjustable attack magnitude

To determine the value of $\beta$, UA-RL uses the Nelder-Mead method Nelder & Mead (1965), which is a direct search method for nonlinear optimization problems, where the derivative is not required. At each iteration, UA-RL uses the accuracy difference between two FL iterations as a reward. More clearly, the reward is defined as $reward_t = accuracy_{t-1} - accuracy_t$, where the $accruacy_{t-1}$ and $accruacy_t$ are respectively the accuracy computed from the server models, $w^{t-1}$ and $w^t$ on the database pooled from all malicious clients' local datasets. When the $reward_t$ is negative, it means that the action is effective. The action is a real number, whose domain is from negative infinity to positive infinity. The action value determines the attack magnitude value $\beta$ to perform the attack. UA-RL collects the corresponding reward and action pairs from the current and previous iterations. These reward and action pairs are used to determine a new action value for the next iteration. More clearly, the Nelder-Mead method takes two reward and action pairs to determine the action for the next attack, i.e.,

$$\text{action}_{t+1} = \varphi\Big((\text{action}_t, \text{reward}_t), (\text{action}_{t-1}, \text{reward}_{t-1})\Big). \tag{1}$$

where $\varphi$ represents the Nelder-Mead method, and $action_t$ represents the action at the $i^{th}$ iteration. Since the searching dimension for $\beta$ is one dimension, the Nelder-Mead method only requires two reward and action pairs from the current and previous iterations. We use action and reward pairs, instead of $\beta$ and reward pairs as inputs to the Nelder-Mead method. This is due to the range and domain of the Nelder-Mead method being both from negative infinity to positive infinity for this one-dimensional problem, but $\beta > 0$. Note that the Nelder-Mead method takes one action point and a predefined parameter, called step to start the search. More clearly, no reward is needed in the first two Nelder-Mead method iterations. Once $action_{t+1}$ is obtained from the Nelder-Mead method, UA-RL maps the action value to $\beta$ through a modified Sigmoid function,

$$\beta_t = \frac{z_a}{1 + e^{-z_b(action_t - z_c)}} \tag{2}$$

where $z_a$, $z_b$ and $z_c$ are hyper-parameters. $z_a$ controls the range of the function such that $\beta \in (0, z_a)$. $z_b$ controls the shape of the function, and $z_c$ shifts the action. Once $\beta_{t+1}$ is obtained, it is used to determine $\widetilde{w}_a^t = w^t - \beta_{t+1}\Psi_a^t$, which is the poisoned model that sent back to the server.

In the first $d$ iterations, UA-RL does not use the Nelder-Mead method to perform the search. It randomly selects a node in a predefined list as an action and maps this action to $\beta$ to perform that attack. Then, the reward for this action is recorded in the next iteration. In the next iteration, another action in $E$ is randomly selected to perform the attack. After the first $d$ iterations, the Nelder-Mead method is used to determine the action for the attack. However, if the rewards are consistently greater than zero in $R$ iterations, a point within a predefined central domain is randomly selected as an action to perform the attack. The pseudo-code of UA-RL is given in Algorithm 1.

## 4 EXPERIMENTS AND RESULTS

### 4.1 EXPERIMENTS SETUPS

We evaluate the proposed UA-RL on three datasets, MINST, Fashion-MINST, and EMINST, which are widely employed in FL studies[1]. Following the previous work in Awan et al. (2021), we use the Dirichlet distribution to control the heterogeneity of the data distribution, where the Dirichlet $\alpha \in [0.2, 1.0, 20]$. The lower the $\alpha$ the more unevenly the data is distributed between clients by class label. In all the experiments, the total number of clients was 40. We set the malicious population

---

[1]We will make our code public after the publication.

---

**Algorithm 1** Untargeted attack algorithm based on reinforcement learning (UA-RL)

---

**Given**:
- $E$ (empirical points): a list of length $d$
- $R$ (restart threshold): an integer
- $C$ (central domain): a real interval
- $s$: the step size for the Nelder-Mead Search algorithm

**Malicious Client Group:**

*Initialization*:

**for** $t = 0$ to $d - 1$ **do**
    Receive $w^t$ from the Server
    Record reward for action in previous round $action_{t-1}$ when $t \geq 1$
    Randomly select $action$ in $E$. Each $action$ can only be selected once.
    Proceed with local training and obtain $w_a^t$
    Send the poisoned model $\widetilde{w}_a^t = w^t - \beta \left( w_a^t - w^t \right)$ back to the Server
**end for**
Pick the node in $E$ with the best reward as $e$. Start the *Nelder-Mead search phase* with $(e, s)$

*Nelder-Mead search phase*:

**for** $t = d$ to $T$ **do**
    Receive $w^t$ from the Server
    Record reward for action in previous iteration $action_{t-1}$
    **if** in $R$ iterations, the rewards are greater than 0 **then**
        Generate a random number $c$ in $C$. Restart the *Nelder-Mead search phase* with $(c, s)$
    **else**
        Determine $action_t$ using the Nelder-Mead method and previous rewards and actions
    **end if**
    Compute $\beta$ using Eq.2
    Proceed with local training and obtain $w_a^t$
    Send the poisoned model $\widetilde{w}_a^t = w^t - \beta \left( w_a^t - w^t \right)$ back to the Server
**end for**

---

as 20% and 40% of the whole FL clients. Due to the randomness of the algorithm, each experiment runs five times, and the average is reported as the final result.

The aggregation rules tested in these experiments are FedAvg, CONTRA, FoolsGold, geometric median, and Krum. We chose the geometric median and Krum because of their robustness. They were also tested in the previous FL attack studies Xie et al. (2020) Baruch et al. (2019) Pillutla et al. (2022). We also include CONTRA and FoolsGold, because their design principles are different from the geometric median and Krum. CONTRA, FoolsGold, geometric median, and Krum are state-of-the-art aggregation algorithms designed for robust FL to defend against different attacks from clients. We chose FedAvg, which has no defense feature, as a reference.

Since UA-RL is the first method developed to perform a black-box attack against FL, no previous works can be employed for direct comparison. We use two other attack methods as our baselines: fixed reverse gradient (FR) and untargeted label flipping (UT). Fixed reverse gradient adopts a fixed $\beta = 1$ without the ability for adjustment. Untargeted label flipping trains the malicious models with a mismatched mapping of the class label and the training data, with $\{0 \rightarrow 1, 1 \rightarrow 2, \cdots, 0 \rightarrow 9\}$. We also include the training without attack (NO) as a comparison. In the following experiments, $z_a$, $z_b$, and $z_c$ are set to 2, -0.7, and 0 respectively.

## 4.2 RESULTS

Table 1 shows the results of MNIST. The proposed UA-RL can significantly deteriorate the training process in most of the cases and on average, it can reduce the accuracy by 30.03%. Compared with the baselines, RF, and UT, UA-RL achieves the best performance. Among the 30 comparisons in Table 1, UA-RL achieves 19 lowest accuracies, while FR and UT achieve 5 and 6 lowest accuracies, respectively. We also note that all the accuracies of FoolsGold and CONTRA under attacks are very

low. To further evaluate CONTRA and FoolsGold, we reduce the malicious population to 2, which is 5% of the total population, and attack them with UA-RL. The results are listed in Table 2. Their performance is still unsatisfactory for all three datasets. Krum is relatively robust when $\alpha = 20$, where the class labels are more evenly distributed, and each client has similar label distributions. However, when $\alpha$ becomes small, its UA-RL accuracy drops a lot. The geometric median is the most robust one against the attacks. However, the accuracy of the geometric median also reduces significantly, when the malicious population is 40% and $\alpha = 0.2$ and $\alpha = 1.0$. When the malicious population is only 20%, UA-RL can operate only on $\alpha = 0.2$ for the geometric median, where the label distributions in different datasets have more difference. It should be highlighted that although the geometric median is the most robust aggregation rule against the attacks, its clean accuracy (NO) is 13-15% lower than Contra, FedAvg, and FoolsGold for $\alpha = 0.2$. It is a trade-off between robustness and clean accuracy in uneven label distributions.

Table 1: A comparison of attack performance on different aggregation rules on the MNIST dataset. The numbers in the table are accuracy (%)

| Mali | Aggr rule | $\alpha$ | | 0.2 | | | | 1.0 | | | | 20 | | |
|---|---|---|---|---|---|---|---|---|---|---|---|---|---|
| | | NO | FR | UA-RL | UT | NO | FR | UA-RL | UT | NO | FR | UA-RL | UT |
| 40% | Contra | 70.84 | 23.33 | 22.96 | **3.31** | 78.84 | 31.02 | 9.86 | **6.19** | 79.60 | 15.31 | **5.43** | 5.18 |
| | Fedavg | 68.01 | 14.06 | **13.33** | 27.29 | 78.90 | 31.24 | **11.91** | 27.74 | 80.16 | 35.39 | **9.86** | 20.05 |
| | Foolsgold | 69.83 | 9.80 | **9.63** | 13.61 | 74.82 | 10.32 | **6.18** | 8.47 | 79.61 | 28.20 | 5.75 | **5.62** |
| | Geo med | 55.03 | 20.09 | **16.09** | 44.10 | 78.70 | 41.88 | **33.20** | 70.72 | 79.66 | **73.18** | 74.00 | 80.34 |
| | Krum | 11.61 | **10.28** | 13.31 | 18.13 | 44.58 | 52.79 | **13.48** | 65.12 | 78.97 | 76.77 | **34.60** | 79.22 |
| | **Avg** | 65.93 | 16.82 | **15.50** | 22.08 | 71.17 | 33.45 | **14.93** | 35.65 | 79.60 | 45.77 | **25.93** | 38.08 |
| 20% | Contra | 70.84 | 11.11 | **10.60** | 16.20 | 78.84 | 36.10 | 8.86 | **8.39** | 79.60 | 9.95 | **6.57** | 5.83 |
| | Fedavg | 68.01 | 33.72 | 42.88 | 53.94 | 78.90 | 61.69 | **55.11** | 66.09 | 80.16 | 73.76 | **70.66** | 73.67 |
| | Foolsgold | 69.83 | 25.63 | **11.01** | 13.69 | 74.82 | 42.63 | **7.28** | 8.48 | 79.61 | 11.29 | 6.08 | **5.39** |
| | Geo med | 55.03 | 34.92 | 37.87 | 52.88 | 78.70 | **70.25** | 73.58 | 78.65 | 79.66 | 79.42 | **78.54** | 80.31 |
| | Krum | 11.61 | 18.68 | 10.56 | **10.19** | 44.58 | 66.14 | **39.26** | 40.48 | 78.97 | 78.81 | **77.91** | 79.78 |
| | **Avg** | 65.93 | 26.35 | **25.59** | 34.18 | 71.17 | 55.36 | **36.82** | 40.42 | 79.60 | 50.65 | **47.95** | 49.00 |

[*] We do not include results in Avg when the accuracy of no attack (NO) is lower than 30%.

Table 2: The performance of UA-RL on FoolsGold and CONTRA with 5% malicious population and $\alpha = 20$. The numbers in the table are accuracy (%)

| Aggr rule | EMNIST | F-MNIST | MNIST |
|---|---|---|---|
| Contra | 3.94 | 22.39 | 10.64 |
| Foolsgold | 0 | 0.8 | 9.74 |

We present the averaged UA-RL's action value and the accuracy of MNIST in Table 3. The action values and the accuracy values in the table are average action values and accuracy from the last FL iteration of the 5 experiments on MNIST. As a result, UA-RL selects different action values for different aggregation rules. For CONTRA, FedAvg, and FoolsGold, the action values are positive and relatively large, but for geometric median and Krum, the action values are much smaller. In other words, UA-RL uses a high attack magnitude, $\beta$ to attack CONTRA, FedAvg, and FoolsGold, but uses a low attack magnitude to attack geometric median and Krum. Figure 2 shows the action values and accuracy from all five experiments for each aggregation rule. When $\alpha = 20$, UA-RL uses more similar actions to attack the same aggregation rule in the five experiments. When $\alpha = 1$ and $\alpha = 0.2$, each client has more different label distributions in the 5 experiments, UA-RL selects more diverse actions to perform the attack. The outliers in Figure 2 show that UA-RL may not achieve a successful attack in every experiment, but overall, it achieves the best performance compared to the baselines. Due to the limited space, we provide experimental results of the other two datasets, Fashion-MINST, and EMINST in Appendix A.4.

We can see that UA-RL does not have enough strength to attack FedAvg when the malicious population is 20% and $\alpha = 1$ and 20. (see in Figure 2. When $z_a$ increases to 8, UA-RL can successfully attack FedAvg in all the cases. The experimental results using $z_a = 8$ are given in Appendix A.4.3.

Figure 3 illustrates how the Nelder-Mead search actually works between iterations on the geometric median. In successful attacks, usually after a few iterations, UA-RL moves the action value to

Table 3: UA-RL's averaged action and accuracy records on MNIST. The action number is float and the accuracy number in (%)

| Mali | Aggr rule | Action (α=0.2) | Accuracy (α=0.2) | Action (α=1.0) | Accuracy (α=1.0) | Action (α=20) | Accuracy (α=20) |
|---|---|---|---|---|---|---|---|
| 40% | Contra | 1.3 | 22.96 | 5.0 | 9.86 | 3.0 | 5.43 |
| | Fedavg | 9.5 | 13.33 | 5.4 | 11.91 | 4.8 | 9.86 |
| | Foolsgold | 3.0 | 9.63 | 5.0 | 6.18 | 1.5 | 5.75 |
| | Geo med | -4.0 | 16.09 | 0.5 | 33.20 | -0.6 | 74.00 |
| | Krum | 1.4 | 13.31 | -1.6 | 13.48 | -3.7 | 34.60 |
| 20% | Contra | 4.4 | 10.60 | 4.6 | 8.86 | 3.2 | 6.57 |
| | Fedavg | 5.1 | 42.88 | 3.5 | 55.11 | -0.9 | 70.66 |
| | Foolsgold | 3.2 | 11.01 | 5.2 | 7.28 | 2.6 | 6.08 |
| | Geo med | -0.6 | 37.87 | -0.8 | 73.58 | -0.9 | 78.54 |
| | Krum | -2.2 | 10.56 | -1.6 | 39.26 | -0.2 | 77.91 |

one direction, which stops FL from convergence. While in unsuccessful attempts, the UA-RL uses action values in a wide range, but the effect is not obvious. The experimental results for the other four aggregation rules are given in Appendix A.4.4.

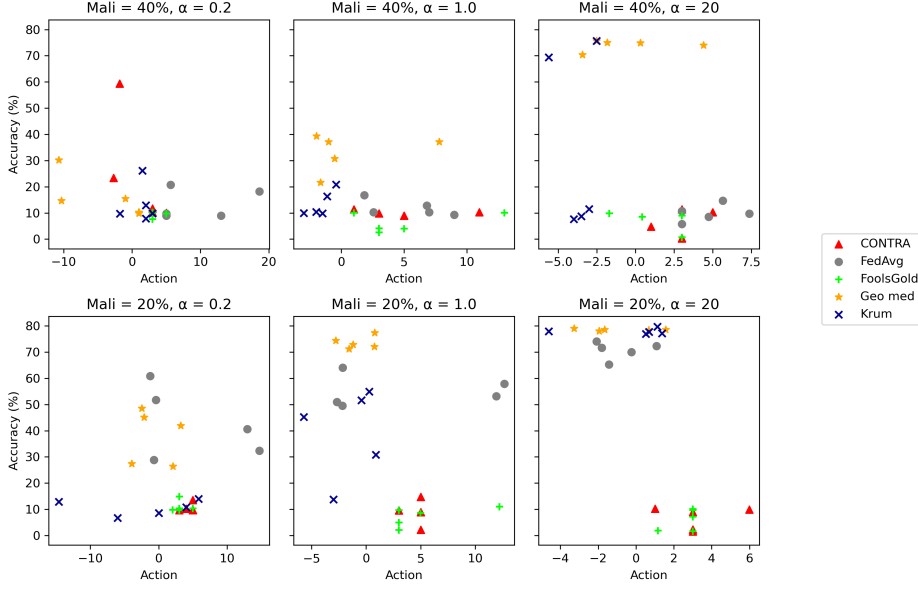

Figure 2: UA-RL action vs accuracy on MNIST

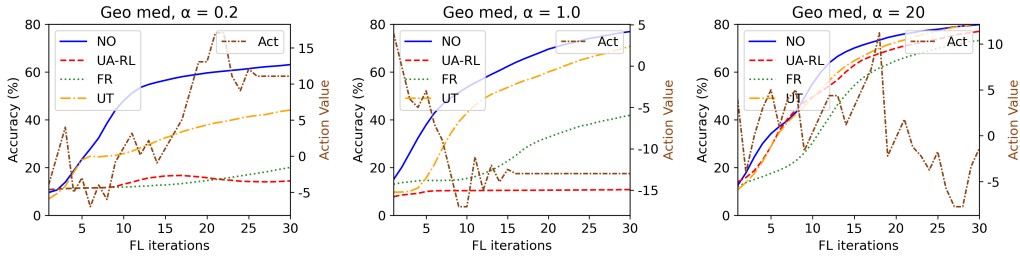

Figure 3: Four aggregation rule's validation history on MNIST with action value of UA-RL. 40% malicious population on geometric median

## 5 DISCUSSION AND ANALYSIS

In this section, we analyze the observations from the experimental results theoretically. We mainly focus on Krum and FoolsGold, because the weakness of FedAvg is well-known Blanchard et al. (2017); CONTRA is very similar to FoolsGold, and the robustness of the geometric median has been well-analyzed by previous works Minsker (2015) Wu et al. (2020). From the experiment, we observe that (*i*) the performance of FoolsGold deteriorates significantly in all the data distributions and percentage of malicious clients and (*ii*) Krum can defend the proposed attack under $\alpha = 20$, which is close to i.i.d. datasets and 20% malicious clients. We also observe that (*iii*) UA-RL exploits different attack magnitudes to defeat Krum and FoolsGold. More precisely, UA-RL uses a low attack magnitude against Krum, but a large magnitude against FoolsGold. Appendix A.3 provides the algorithms of Krum, FoolsGold and geometric median.

### 5.1 ANALYSIS OF KRUM

Krum uses the sum of local gradient vectors $V_i^t$ to perform the updates i.e., $V_i^t = w^t - w_i^t$. To simplify the notation, the superscript $t$ on $V_i^t$ is ignored. In this analysis, there are $n-1$ honest clients and one malicious client. Their corresponding sum of local gradient vectors are $S_o = \{V_1 \cdots V_{n-1}\}$ and $V_a$, respectively. We denote $S_u = \{V_1, \cdots, V_{n-1}, V_a\}$. The malicious client sets their attack vector as $V_a = -\beta V_n$, where $V_n$ is the sum of local gradient vectors from their original dataset and $\beta > 0$. An attack is considered a success at an iteration if $V_a$ is selected. Krum uses the score function $s(i) = \sum_{j=1}^{n-f-2} \left\| V_i - V_{i(j)} \right\|^2$, where $f$ is a predefined constant and $V_{i(j)}$ is one of the $n - f - 2$ closest vectors to $V_i$, to assign score to each vector. The vector with the lowest score is selected by Krum. Let $s_o(m^*) < s_o(i)$, $\forall i \neq m^*$ and both $m^*$ and $i \in \{1, \ldots, n-1\}$. $s_o(m^*)$ is only computed from the $n - f - 2$ closest honest vectors to $V_{m^*}$. In other words, $s_o(m^*)$ is the minimum score from $S_o$, the set of honest clients' vectors. Similarly, $s_o(i)$ is the score of $V_i$ and $s_o(i)$ is computed from the $n - f - 2$ closest honest clients' vectors to $V_i$. We use $s_u(i)$ to denote that the score of $V_i$, when it is computed from the set $S_u$, including the malicious client's vector $V_a$ and $\theta_{i(j)}$ to denote the angle between $V_i$ and $V_{i(j)}$. The following theorem indicates that under certain conditions, Krum can be broken by a single attacker.

**Theorem 1** *If one of the following conditions is true, $\ni \beta > 0$ such that $-\beta V_n$ can successfully attack Krum.*

*(i)* $s_o(m^*) < s_u(i)$, $\forall i \neq m^*$; *both $m^*$ and $i \in \{1, \ldots, n-1\}$ and*

$$\|V_{m^*}\| > \frac{2}{n-f-2} \sum_{j=1}^{n-f-2} \|V_{m^*(j)}\| \cos\left(\theta_{m^*(j)}\right), \tag{3}$$

*where all $V_{m^*(j)}$ are the honest clients' vectors.*

*(ii)* $s_u(e^*) < s_o(m^*)$ *and*

$$(n-f-2)\|V_{e^*}\|^2 - 2\|V_{e^*}\| \sum_{j=1}^{n-f-3} \|V_{e^*(j)}\| \cos\left(\theta_{e^*(j)}\right) - \|V_z\|^2 > 0, \tag{4}$$

*where $e^*$ is an honest client, $s_u(e^*) = \min_{i \in \{1, \cdots, n-1\}} s_u(i)$, and $V_z \in S_o$ but $V_z \neq V_{e^*(q)}$.*

The proof is given in the Appendix A.2.1. Theorem 1 only mentions that there exists a $\beta$ that can successfully attack Krum if the conditions hold. The proof in Appendix A.2.1 in fact shows that a smaller $\beta$ is more suitable for Eq.3 and $\beta$ is requested in a particular range for Eq.4. Please see the details in the proof. If $\beta$ is too small, the attack is not effective. Thus, UA-RL searches different $\beta$ to perform the attack. Although in this analysis, one malicious client may be enough to perform a successful attack, in practice, it is not. Because the above analysis only considers one iteration and a small $\beta$ may require performing an attack. As a result, many iterations are needed. Since $V_i$ changes in each iteration, the conditions may not hold for every iteration. To compromise Krum, in practice, more malicious clients may be needed.

To understand why Krum is robust under an i.i.d. setting, we further analyze the conditions (*i*) and (*ii*) in Theorem 1. Corollary 1 pinpoints that under the assumptions: $\cos\left(\theta_{m^*(j)}\right)$ and $\left\|V_{m^*(j)}\right\|$ are independent; $\|V_i\|$ is independent; $E\left(\|V_i\|\right) = \mu$, and $\cos\left(\theta_{m^*(j)}\right) > 0.5$, $\forall m^*\left(j\right)$, the inequality in Eq.3 cannot be valid and the probability of $s_u\left(e^*\right) < s_o\left(m^*\right)$ is very low if $f + 2$ is large. The assumptions are more valid in the i.i.d. datasets, and therefore, Krum is more robust against UA-RL attacks in the i.i.d. datasets. It should be highlighted that $\theta_{m^*(j)}$ is the angle between $V_{m^*}$ and $V_{m^*(j)}$. $\left\|V_{m^*(j)}\right\|$ is the magnitude of $V_{m^*(j)}$ only. Based on the i.i.d. assumption, the independent assumption of $\cos\left(\theta_{m^*(j)}\right)$ and $\left\|V_{m^*(j)}\right\|$ is valid.

**Corollary 1** *If $\cos\left(\theta_{m^*(j)}\right)$ and $V_{m^*(j)}$ are independent; $\|V_i\|$ is independent; $E\left(\|V_i\|\right) = \mu$, and $\cos\left(\theta_{m^*(j)}\right) > 0.5$, $\forall m^*\left(j\right)$, then the inequality in Eq.3 cannot be valid and the probability of $s_u\left(e^*\right) < s_o\left(m^*\right)$ is smaller than $(1/2)^{f+2}$.*

## 5.2 ANALYSIS OF FOOLSGOLD

From the experiments, we observe that UA-RL deteriorates the performance of FoolsGold significantly in all the data distributions and all the percentages of malicious clients. It seems that FoolsGold is very vulnerable to UA-RL attacks, and UA-RL uses a relatively large attack magnitude to attack FoolsGold. In the following analysis, we attempt to explain these observations theoretically.

Let $H_i$ be the aggregate historical features i.e., $H_i = \sum_{t=1}^{T} V_i^t$, $cs_{ij} = \cos\left(H_i, H_j\right)^2$, and $v_i = max_j(cs_{ij})$. FoolsGold does not compute $cs_{ii}$ for all $i$. It rescales all $cs_{ij}$ by using $cs_{ij}' = cs_{ij} \times v_i/v_j$ if $v_j > v_i$ and then computes $\alpha_i = 1 - max_j(cs_{ij}')$. It normalizes $\alpha_i$ by using $\alpha_i' = \alpha_i/\max(\alpha)$. The final weights are $a_i = \kappa\left(\ln\left(\frac{\alpha_i'}{1-\alpha_i'}\right) + 0.5\right)$ and the Federated SGD uses $w_T = w_{T-1} + \sum_{i=1}^{n} a_i V_{i,T}$ to perform the updates. Theorem 2 indicates that FoolsGold uses only the information from the original datasets to compute $a_i$, the weights used to aggregate $V_i^T$ to update the server model weights, and neglects all the information that can separate the malicious clients from honest clients.

**Theorem 2** *If there are two malicious clients, who have their own datasets; the attack magnitude $\beta > 0$ is set as a constant, and the cosine similarity of any two historical features from the original datasets, including those owned by malicious clients, is positive, then FoolsGold does not use any information that can distinguish the honest and malicious clients to compute $a_i$, the weights used to aggregate $V_i^T$ to update the server model weights.*

Based on Theorem 2, FoolsGold would consider the two malicious clients as honest ones. If $\beta$ is very large, the malicious clients can take over the weighted sum i.e., $\sum_{i=1}^{n-2} a_i V_i^t - \beta \sum_{i=n-1}^{n} a_i V_i^t$, in the update equation. In other words, two malicious clients are enough to perform an effective attack. The result in Table 2 demonstrates that two malicious clients are enough to stop the learning process of FoolsGold, which matches our theoretical analysis. We also can see from Table 1 and Figure 2 that UA-RL uses larger action values, implying larger $\beta$, to perform the attacks.

## 6 CONCLUSION

In this study, we performed the first untargeted black-box attack against the FL system using robust aggregation rules. We proposed the UA-RL attack, which uses reinforcement learning to search for a suitable attack magnitude to perform the attack. The experimental results show that UA-RL effectively attacks most of the aggregation rules tested in the experiments in a non-i.i.d. setting. The experiments show that FoolsGold and CONTRA are very vulnerable to UA-RL attacks. Krum is only robust when the datasets are i.i.d. and the population of malicious clients is low, 20% in the experiment. We also found that UA-RL uses different attack magnitudes to attack different aggregation rules. In addition to the experiments, we offered theories to explain the observations.

---

[2]In FoolsGold, feature weighted cosine similarity is used. Without lost generality, cosine similarity is used in this analysis.

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

# A APPENDIX

## A.1 NOTATIONS

Please find the notations used in this paper summarized in Table 4.

Table 4: Notations

| PARAMETER | DESCRIPTION |
| --- | --- |
| $n$ | total number of FL clients |
| $m$ | number of malicious clients |
| $action$ | action of the UA-RL |
| $Aggr$ | server's Aggregation rule |
| $\beta$ | attack magnitude |
| $J$ | number of epochs in local training |
| $\Psi_a^t$ | the sum of negative gradients of the client $a$ |
| $w^t$ | the server's model at iteration $t$ |
| $w_i^t$ | the model of malicious client $i$ at iteration $t$ |

## A.2 MATHEMATICAL PROOFS

### A.2.1 PROOF OF THEOREM 1

**Case 1:** According to the assumption, $s_o(m^*) < s_u(i), \forall i \neq m^*$ and $m^*$ $and$ $i \in \{1, \ldots, n-1\}$, the malicious clients' vector $V_a$ does not influence the original minimum score among honest workers. Now, we consider, $s_o(m^*) - \psi(a)$, where $\psi = \sum_{j=1}^{n-f-2} \left\| V_a - V_{m*(j)} \right\|^2$.

$$s_o(m^*) - \psi(a) = \sum_{j=1}^{n-f-2} \left\| V_{m^*} - V_{m^*(j)} \right\|^2 - \left\| V_a - V_{m^*(j)} \right\|^2$$

$$= \sum_{j=1}^{n-f-2} \left\| V_{m^*} \right\|^2 - 2 \left\| V_{m^*} \right\| \left\| V_{m^*(j)} \right\| \cos\left(\theta_{m^*(j)}\right) - \left\| V_a \right\|^2 + 2 \left\| V_a \right\| \left\| V_{m^*(j)} \right\| \cos\left(\theta_{a,m^*(j)}\right)$$

where $\theta_{a,\,m^*(j)}$ is the angle between $V_a$ and $V_{m^*(j)}$. Substituting $V_a = -\beta V_n$ and grouping the terms, we have:

$$= (n - f - 2) \left\| V_{m^*} \right\|^2 - 2 \left\| V_{m^*} \right\| \sum_{j=1}^{n-f-2} \left\| V_{m^*(j)} \right\| \cos\left(\theta_{m^*(j)}\right) \tag{5}$$

$$- \beta \left\| V_n \right\| \sum_{j=1}^{n-f-2} \left( \beta \left\| V_n \right\| + 2 \left\| V_{m^*(j)} \right\| \cos\left(\theta_{n,m^*(j)}\right) \right) \tag{6}$$

According to the assumption, $(n - f - 2) \left\| V_{m^*} \right\|^2 - 2 \left\| V_{m^*} \right\| \sum_{j=1}^{n-f-2} \left\| V_{m^*(j)} \right\| \cos\left(\theta_{m^*(j)}\right) > 0$. Thus, $\exists \beta > 0$ such that $s_o(m^*) - \psi(a) > 0$, meaning that the attack success.

**Case 2:** Since $s_u(e^*) < s_o(m^*)$, it implies $V_a$ is one of the $n - f - 2$ closest vectors to $V_{e^*}$. Let

$$s_u(e^*) = \sum_{j=1}^{n-f-3} \left\| V_{e^*} - V_{e^*(j)} \right\|^2 + \left\| V_{e^*} - V_a \right\|^2 \tag{7}$$

and

$$\phi(a) = \sum_{j=1}^{n-f-3} \left\| V_a - V_{e^*(j)} \right\|^2 + \left\| V_a - V_z \right\|^2 \tag{8}$$

where $V_z \in S_o$ but $V_z \neq V_{e^*(j)}$. Considering $s_u(e^*) - \phi(a)$ and simplifying it, we have

$$\sum_{j=1}^{n-f-3} \left( \left\| V_{e^*} \right\|^2 - 2 \left\| V_{e^*} \right\| \left\| V_{e^*(j)} \right\| \cos\left( \theta_{e^*(j)} \right) - \left\| V_a \right\|^2 + 2 \left\| V_a \right\| \left\| V_{e^*(j)} \right\| \cos\left( \theta_{a,e^*(j)} \right) \right)$$

$$+ \left\| V_{e^*} \right\|^2 - \left\| V_z \right\|^2 - 2 \left\| V_{e^*} \right\| \left\| V_a \right\| \cos\left( \theta_{a,e^*} \right) + 2 \left\| V_z \right\| \left\| V_a \right\| \cos\left( \theta_{a,z} \right)$$

$$= -(n - f - 3) \left\| V_a \right\|^2 + 2 \left\| V_a \right\| \left( \left\| V_z \right\| \cos\left( \theta_{a,z} \right) - \left\| V_{e^*} \right\| \cos\left( \theta_{a,e^*} \right) + \sum_{j=1}^{n-f-3} \left\| V_{e^*(j)} \right\| \cos\left( \theta_{a,e^*(j)} \right) \right)$$

$$+ (n - f - 2) \left\| V_{e^*} \right\|^2 - 2 \left\| V_{e^*} \right\| \sum_{j=1}^{n-f-3} \left\| V_{e^*(j)} \right\| \cos\left( \theta_{e^*(j)} \right) - \left\| V_z \right\|^2$$

In terms of $\|V_a\|$, it is a quadratic equation. Let

$$b = \left( \left\| V_z \right\| \cos\left( \theta_{a,z} \right) - \left\| V_{e^*} \right\| \cos\left( \theta_{a,e^*} \right) + \sum_{j=1}^{n-f-3} \left\| V_{e^*(j)} \right\| \cos\left( \theta_{a,e^*(j)} \right) \right), \tag{9}$$

and

$$c = (n - f - 2) \left\| V_{e^*} \right\|^2 - 2 \left\| V_{e^*} \right\| \sum_{j=1}^{n-f-3} \left\| V_{e^*(j)} \right\| \cos\left( \theta_{e^*(j)} \right) - \left\| V_z \right\|^2. \tag{10}$$

$s_u(e^*) - \phi(a)$ can be rewritten as:

$$s_u(e^*) - \phi(a) = -(n - f - 3) \left\| V_a \right\|^2 + 2b \left\| V_a \right\| + c \tag{11}$$

If $s_u(e^*) - \phi(a) > 0$, we obtain

$$\frac{b + \sqrt{b^2 + (n - f - 3)c}}{(n - f - 3)} > \|V_a\| > \frac{b - \sqrt{b^2 + (n - f - 3)c}}{(n - f - 3)}. \tag{12}$$

Since $\|V_a\| > 0$, the inequality change to

$$\frac{b + \sqrt{b^2 + (n - f - 3)c}}{(n - f - 3)} > \|V_a\| > \max\left( 0, \frac{b - \sqrt{b^2 + (n - f - 3)c}}{(n - f - 3)} \right) \tag{13}$$

If $c > 0$, or $b > 0$, $\frac{b + \sqrt{b^2 + (n-f-3)c}}{(n-f-3)} > 0$. As a result, $\exists \beta > 0$ such that $s_u(e^*) - \phi(a) > 0$. Note that $V_a = -\beta V_n$ and $\theta_{a,e^*}$, $\theta_{a,e^*(j)}$ and $\theta_{a,z}$ are the angles between $V_a$ and $V_{e^*}$, $V_{e^*(j)}$ and $V_z$, respectively. Let the angles between $V_n$ and $V_{e^*}$, $V_{e^*(j)}$ and $V_z$ be $\theta_{n,e^*}$, $\theta_{n,e^*(j)}$ and $\theta_{n,z}$. Thus, $\theta_{n,e^*} = \pi - \theta_{a,e^*}$, $\theta_{n,e^*(j)} = \pi - \theta_{a,e^*(j)}$, and $\theta_{n,z} = \pi - \theta_{a,z}$. Since $V_k$, $V_{e^*}$, $V_{e^*(j)}$ and $V_z$ are derived from unmodified data, we assume that $\theta_{n,e^*}$, $\theta_{n,z}$, $\theta_{n,e^*(j)} < \pi/2$ and b can be rewritten as

$$b = \left( -\|V_z\|\cos(\theta_{n,z}) + \|V_{e^*}\|\cos(\theta_{n,e^*}) - \sum_{j=1}^{n-f-3} \|V_{e^*(j)}\|\cos(\theta_{n,e^*(j)}) \right) \quad (14)$$

Thus, $b$ is likely negative and we remove $b > 0$ in the theorem.

$\square$

### A.2.2 PROOF OF COROLLARY 1

Applying the assumptions, $\cos(\theta_{m^*(j)})$ and $\|V_i\|^2$ are independent, $E(\|V_i\|) = \mu$, and $\cos(\theta_{m^*(j)}) > 0.5$, $\forall m^*(j)$ and taking expectation on the left-hand side of Eq.3, we have:

$$\frac{2}{n-f-2} \sum_{j=1}^{n-f-2} E(\|V_{m^*(j)}\|) E(\cos(\theta_{m^*(j)})) > \mu. \quad (15)$$

However, the expectation of $\|V_{m^*}\|$, the right-hand side of Eq.3, is also equal to $\mu$, implying that under the assumptions, Eq.3 cannot be valid.

$s_u(e^*) < s_o(m^*)$ means that $V_a$ is one of the $n - f - 2$ closest vectors to $V_{e^*}$, implying that $\|V_{e^*} - V_a\|^2 < \|V_{e^*} - V_i\|^2$, where $V_i$ are all vectors not belonging to the $n - f - 2$ closest vectors to $V_{e^*}$. Simplifying the inequality, we have:

$$\beta^2 \|V_n\|^2 + 2\|V_{e^*}\| (\|V_i\|\cos(\theta_{e^*,i}) + \beta\|V_n\|\cos(\theta_{e^*,n})) < \|V_i\|^2. \quad (16)$$

Note that $V_a = -\beta V_n$ and $V_n$ are from the malicious client's unmodified dataset. Applying the assumptions, $\cos(\theta_{e^*,i}) > 0.5$ and all vector magnitudes, $\|V_i\|$ are independent, even setting $\beta = 0$, the probability of the inequality being valid is smaller then $(1/2)^{f+2}$ for all $f + 2$ vectors $V_i$, not including in the $n - f - 2$ closest vectors to $V_{e^*}$. When $f + 2$ is large, this probability is very low.

$\square$

### A.2.3 PROOF OF THEOREM 2

Assume that there are $n$ clients. The first $n - 2$ clients are honest and the last two clients are malicious. The malicious clients have their own databases, and their corresponding unmodified updates are $V_{n-1}^t$ and $V_n^t$. The malicious clients use $-\beta V_{n-1}^t$ and $-\beta V_n^t$, where $\beta > 0$ to perform the attack.

To simplify the analysis, $\beta$ is set as a constant. Historical features from the malicious clients are $H_{n-1} = -\beta \sum_{t=1}^{T} V_{n-1}^t$ and $H_n = -\beta \sum_{t=1}^{T} V_n^t$. Assume that the cosine similarity of any two historical features from the original datasets, including those owned by malicious clients, is positive. Based on this assumption, $cs_{ij} > 0, \forall 1 \le i, j \le n - 2$ or $\forall n - 1 \le i, j \le n$; otherwise $cs_{ij} < 0$. It should be highlighted that $cos(H_{n-1}, H_n) = cos(\sum_{t=1}^{T} V_{n-1}^t, \sum_{t=1}^{T} V_n^t)$. It means that $cs_{n-1} = cos(H_{n-1}, H_n)$ cannot be used to detect the attack, because it only reveals information from the original datasets.

Note that for $1 \le i \le n - 2$ and $n - 1 \le j \le n$, $cs_{i,j} = \cos\left(\sum_{t=1}^{T} V_i^t, -\beta \sum_{t=1}^{T} V_j^t\right) = -\cos\left(\sum_{t=1}^{T} V_{i,t}, \sum_{t=1}^{T} V_{j,t}\right) < 0$. They expose the malicious clients' information. All $v_i =$

$max_j(cs_{ij})$ are positive because each row has some positive values. Furthermore, $cs_{n,n-1} = cs_{n-1,n} = v_n = v_{n-1}$ because FoolsGold does not compute $cs_{n,n}$ and $cs_{n-1,n-1}$ and all other $cs_{n-1,j}$ and $cs_{n,j}$, where $1 \leq j \leq n-2$ are negative. Fig. 4 illustrates $cs_{ij}$ and $v_i$. Since all $v_i$ are positive, the signs of $cs_{ij}'$ and $cs_{ij}$ are the same. Since $\frac{v_i}{v_j} < 1, cs_{ij}' \leq cs_{ij}, \forall i, j$. Since $v_n = v_{n-1}, cs_{n-1,n}' = cs_{n,n-1}' = cs_{n,n-1} = cs_{n-1,n}$. In other words, the rescaling is not applied to $cs_{n,n-1} = cs_{n-1,n}$. Clearly, for the first $n-3$ rows, $max_j(cs_{ij}')$ is from the first $n-3$ columns and for the last two rows, $max_j(cs_{ij}') = cs_{n,n-1} = cs_{n-1,n}$ are from the two malicious clients. We can obverse that for $n-1 \leq i \leq n, \alpha_i = 1 - max_j(cs_{ij}')$ do not use any information from the honest clients and for $1 \leq i \leq n-2, \alpha_i = 1 - max_j(cs_{ij}')$ do not use any information from the malicious clients.

The last two steps $\alpha_i' = \alpha_i / \max(\alpha)$ and $a_i = \kappa \left( \ln \left( \frac{\alpha_i'}{1-\alpha_i'} \right) + 0.5 \right)$ normalize and non-linearly map $\alpha_i'$ to the weight $a_i$. Since $cs_{n,n-1} = cs_{n-1,n} = cos(H_{n-1}, H_n) = cos(\sum_{t=1}^T V_{n-1}^t, \sum_{t=1}^T V_n^t)$ and $\alpha_i'$ only use information from $cs_{i,j} > 0$, which are from the original datasets, FoolsGold does not use any information that can distinguish honest and malicious clients to compute $a_i$. If $\beta$ is very large, the malicious clients can take over the weighted sum i.e., $\sum_{i=1}^{n-2} a_i V_i^t - \beta \sum_{i=n-1}^n a_i V_i^t$, in the update equation. In other words, two malicious clients are enough to perform the attacks. $\square$

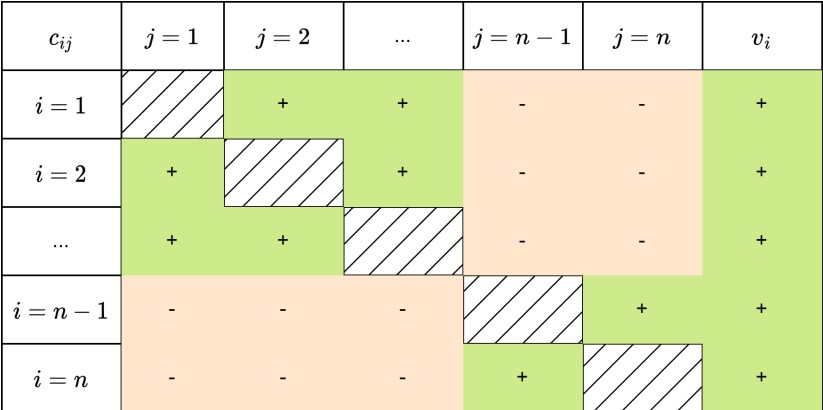

Figure 4: FoolsGold's cosine similarities between clients

### A.3 ALGORITHMS

#### A.3.1 FEDAVG

For FedAvg, the aggregated rule, $Aggr$ in Algorithm 2 is

$$\frac{1}{n} \times \sum_{i=1}^n w_i^t \tag{17}$$

#### A.3.2 KRUM

Krum selects a single client in every FL iteration. The chosen client is the one with parameters that are closest to another $n - f - 2$ clients:

$$\text{Krum}(\mathcal{P}) = \left( p_i \mid \text{argmin}_{i \in [n]} \sum_{i \to j} \|p_i - p_j\|^2 \right), \tag{18}$$

where $i \to j$ is the $n - f - 2$ nearest neighbors to $p_i$ in $P$, measured by Euclidean Distance.

---

**Algorithm 2** Federated Learning algorithm

---

**Server:**
Initialization $w^0 \leftarrow random()$
**for** $t = 0$ to $T$ **do**
    Broadcast $w^t$ to all $n$ clients
    Wait for clients' local training process
    Receive the weights, $w_i^t$ from $n$ clients
    Computer the aggregated weight: $\bar{w}^t = Aggr(w_1^t \cdots w_n^t)$
    Update $w^{t+1} = w^t - lr_{server}(w^t - \bar{w}^t)$, where $lr_{server}$ is server's learning rate
**end for**

**Clients:** $i = 1, ..., n$
**for** $t = 0$ to $T$ **do**
    Receive $w^t$ from the Server
    $w_i^0 = w^t$
    **for** $j = 0$ to $J - 1$ **do**
        Perform training with SGD: $w_i^{j+1} = w_i^j - lr_{local} \times \nabla F(w_i^j)$, where $F$ is a loss function
        and $lr_{local}$ is a local learning rate
    **end for**
    Set $w_i^t = w_i^J$
    Send local weights $w_i^t$ to Server
**end for**

---

### A.3.3  GEOMETRIC MEDIAN

The geometric median of $\{y_1, \cdots, y_n\}$, denoted by $med\{y_1, \cdots, y_n\}$, is defined as:

$$\mathrm{med}\,\{y_1, \ldots, y_n\} \triangleq \arg\min_{y \in \mathbb{R}^d} \sum_{i=1}^n \|y - y_i\|.$$

Here, $\mathrm{argmin}$ means the value of the argument $y$ which minimizes the sum.

### A.3.4  FOOLSGOLD

Please see FoolsGold at Algorithm 3.

### A.3.5  CONTRA

CONTRA can be viewed as FoolsGold with a function on the client's history to calculate its reputation scores, which also affects the calculation of the weights for aggregation besides the cosine similarity. Please view the CONTRA algorithm on page 13 of Awan et al. (2021).

---

**Algorithm 3** FoolsGold learning algorithm

---

**Data:** Initial Model $w_0$ and SGD updates $\Delta_{i,t}$ from each client $i$ at iteration $t$. Confidence parameter $\kappa$

**for** $t = 0$ to $T$ **do**
    // Per client learning rate for iteration $T$
    Initialize $\alpha$
    **for** All client from $0$ to $i$ **do**:
        // Updates history
        Let $H_i$ be the aggregate historical vector $\sum_{t=1}^{T} \Delta_{i,t}$
        // Feature importance
        Let $S_T$ be the weight of important features at iteration $T$
        **for** all other client from $0$ to $j$ **do**:
            Let $cs_{ij}$ be the $S_T$-weighted cosine similarity between $H_i$ and $H_j$
        **end for**
        Let $v_i = max_j(cs_i)$
    **end for**
    **for** All client from $0$ to $i$ **do**:
        **for** All client from $0$ to $j$ **do**:
            // Pardoning
            **if** $v_i > v_j$ **then**
                $cs_{ij}* = v_i/v_j$
            **end if**
        **end for**
        // Per-row maximums
        $\alpha_i = 1 - max_j(cs_i)$
    **end for**
    // Normalize learning rates to 0-1 range
    $\alpha = \alpha/max_i(\alpha)$
    // Element-wise logit function
    $\alpha = \kappa(ln[(\alpha)/(1-\alpha)] = 0.5)$
    // Federated SGD iteration
    $w_T = w_{T-1} + \sum_i \alpha_i, \Delta_i$
**end for**

---

A.4    THE FULL EXPERIMENTS RESULTS

A.4.1    PARAMETERS OF THE EXPERIMENTS

The parameters employed in the experiments can be found in Table 5. Two sets of hyper-parameters are shown in Table 6. The experimental results for EMNIST, and Fashion MNIST under Set1 setting are in Table 7, and Table 8. Their UA-RL's action vs accuracy plots are in Figure 5, and Figure 6.

Table 5: The parameters employed in the experiments

| Parameter | MNIST | EMNIST | F-MNIST |
|---|---|---|---|
| Local learning rate | 0.001 | 0.001 | 0.001 |
| Server learning rate | 0.1 | 0.1 | 0.1 |
| Iterations | 30 | 30 | 30 |
| Number of epochs | 5 | 2 | 10 |
| Batch size | 32 | 32 | 32 |
| Momentum | 0.9 | 0.9 | 0.9 |
| CNN model layers | 3 | 7 | 12 |
| $f$ of Krum | 15 | 15 | 15 |

Table 6: The parameters of modified Sigmod function, Eq.2

| | $z_a$ | $z_b$ | $z_c$ | Empirical points | Central domain for restart | Step size |
|---|---|---|---|---|---|---|
| Set1 | 2 | -0.7 | 0 | [-4, 0, 4] | [-4, 4] | 1 |
| Set2 | 8 | -0.7 | 2.75 | [-4, 0, 4] | [-4, 4] | 1 |

### A.4.2 EXPERIMENTS RESULT FOR ALL DATASETS UNDER SET1 SETTING

Table 7: A comparison of attack performance on different aggregation rules on the EMNIST dataset. The numbers in the table are accuracy (%)

| Mali | Aggr rule | 0.2 | | | | 1.0 | | | | 20 | | | |
|---|---|---|---|---|---|---|---|---|---|---|---|---|---|
| | $\alpha$ | NO | FR | UA-RL | UT | NO | FR | UA-RL | UT | NO | FR | UA-RL | UT |
| 40% | Contra | 60.38 | 10.52 | 4.70 | **2.02** | 66.33 | 8.14 | **0.00** | 0.71 | 67.64 | 3.63 | **0.00** | 1.20 |
| | Fedavg | 60.30 | 13.05 | 7.26 | **5.32** | 66.46 | 25.88 | **5.20** | 9.40 | 67.67 | 45.95 | **0.48** | 12.08 |
| | Foolsgold | 60.90 | 19.16 | 3.09 | **1.75** | 66.61 | 28.12 | **0.00** | 0.87 | 67.30 | 6.75 | **0.00** | 1.00 |
| | Geo med | 57.39 | 24.87 | **17.15** | 30.18 | 66.18 | 58.03 | 54.94 | **54.01** | 67.27 | 65.86 | 66.31 | **64.18** |
| | Krum | 23.86 | 18.54 | **5.12** | 28.80 | 58.48 | 55.73 | **15.15** | 54.64 | 67.53 | 67.94 | **64.41** | 67.66 |
| | **Avg** | 59.74 | 16.90 | **8.05** | 9.82 | 64.81 | 35.18 | **15.06** | 23.93 | 67.48 | 38.03 | **26.24** | 29.22 |
| 20% | Contra | 60.38 | 17.15 | 6.11 | **1.88** | 66.33 | 20.15 | **1.40** | 1.49 | 67.64 | 51.79 | 6.38 | **1.08** |
| | Fedavg | 60.30 | 39.48 | **31.02** | 34.27 | 66.46 | 59.59 | 57.37 | **39.97** | 67.67 | 64.24 | 62.91 | **41.57** |
| | Foolsgold | 60.90 | 29.41 | **7.00** | 3.42 | 66.61 | 43.75 | **0.02** | 1.30 | 67.30 | 64.15 | **0.00** | 1.28 |
| | Geo med | 57.39 | 40.90 | 43.48 | 53.40 | 66.18 | 64.89 | **64.48** | 65.03 | 67.27 | 67.20 | 67.57 | **67.26** |
| | Krum | 23.86 | 23.81 | **14.65** | 26.89 | 58.48 | 58.15 | 55.74 | **54.37** | 67.53 | 65.79 | **67.19** | 67.86 |
| | **Avg** | 59.74 | 31.74 | **21.90** | 23.24 | 64.81 | 49.31 | 35.80 | **32.43** | 67.48 | 62.63 | 40.81 | **35.81** |

\* We do not include results in Avg when the accuracy of no attack (*NO*) is lower than 30%.

Table 8: A comparison of attack performance on different aggregation rules on the Fashion-MNIST dataset. The numbers in the table are accuracy (%)

| Mali | Aggr rule | 0.2 | | | | 1.0 | | | | 20 | | | |
|---|---|---|---|---|---|---|---|---|---|---|---|---|---|
| | $\alpha$ | NO | FR | UA-RL | UT | NO | FR | UA-RL | UT | NO | FR | UA-RL | UT |
| 40% | Contra | 78.00 | **10.00** | **10.00** | 10.46 | 77.86 | **2.30** | 8.19 | 7.53 | 77.51 | **0.25** | 10.00 | 26.90 |
| | Fedavg | 78.70 | 31.15 | **13.92** | 30.03 | 80.81 | 33.30 | **12.23** | 21.10 | 78.99 | 61.02 | **8.30** | 21.73 |
| | Foolsgold | 76.80 | **10.00** | 10.16 | 3.44 | 78.61 | 7.86 | **10.00** | 3.36 | 79.06 | 10.00 | **7.22** | 30.64 |
| | Geo med | 76.68 | 43.76 | **32.27** | 70.80 | 79.04 | 73.55 | **63.33** | 80.37 | 78.93 | 75.17 | **74.01** | 77.92 |
| | Krum | 47.21 | 27.64 | **12.29** | 56.33 | 76.81 | 75.30 | **8.80** | 75.06 | 78.76 | 79.12 | **77.67** | 78.85 |
| | **Avg** | 71.48 | 24.51 | **15.73** | 34.21 | 78.63 | 38.46 | **20.51** | 37.48 | 78.65 | 45.11 | **35.44** | 47.21 |
| 20% | Contra | 78.00 | **10.00** | 32.22 | 5.38 | 77.86 | **10.00** | 10.00 | 4.74 | 77.51 | **0.23** | 10.04 | 23.34 |
| | Fedavg | 78.70 | 63.47 | **55.26** | 64.13 | 80.81 | 67.49 | **65.10** | 70.64 | 78.99 | **63.72** | 64.00 | 71.39 |
| | Foolsgold | 76.80 | **10.00** | **10.00** | 18.17 | 78.61 | **10.00** | **10.00** | 16.15 | 79.06 | 1.90 | 8.16 | 7.67 |
| | Geo med | 76.68 | **66.40** | 71.21 | 78.09 | 79.04 | **77.84** | 78.23 | 80.11 | 78.93 | 80.41 | **77.46** | 79.46 |
| | Krum | 47.21 | 41.37 | **27.93** | 44.72 | 76.81 | 79.15 | **76.57** | 78.94 | 78.76 | 78.33 | **78.15** | 76.44 |
| | **Avg** | 71.48 | **38.25** | 39.32 | 42.10 | 78.63 | 48.90 | **47.98** | 50.12 | 78.65 | **44.92** | 47.56 | 51.66 |

\* We do not include results in Avg when the accuracy of no attack (*NO*) is lower than 30%.

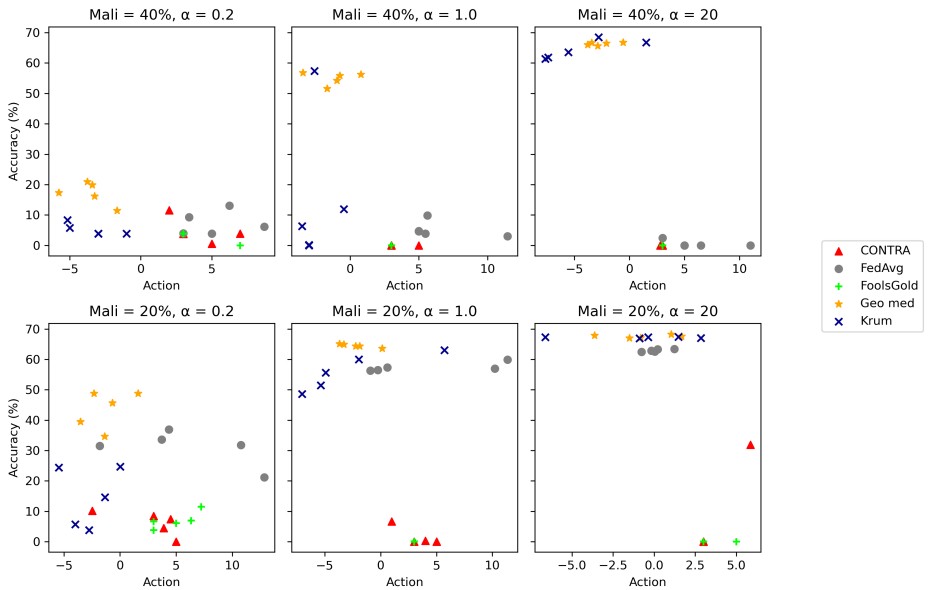

Figure 5: UA-RL action vs accuracy on EMNIST

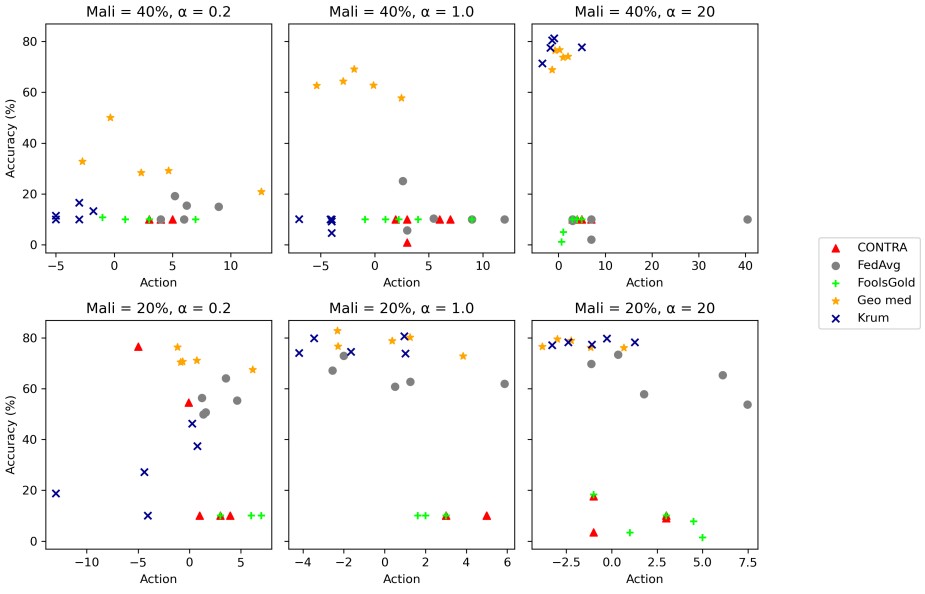

Figure 6: UA-RL action vs accuracy on Fashion-MNIST

### A.4.3 EXPERIMENTS RESULT FOR MNIST UNDER SET2 SETTING

The experimental results for MNIST under Set2 setting are in Table 9, where $z_a = 8$, $z_b = -0.7$, $z_c = 2.75$. This experiment's UA-RL's action vs accuracy is in Figure 7.

Table 9: A comparison of attack performance on different aggregation rules on the MNIST dataset. The numbers in the table are accuracy (%)

| Mali | Aggr rule | $\alpha$ 0.2 NO | FR | UA-RL | UT | 1.0 NO | FR | UA-RL | UT | 20 NO | FR | UA-RL | UT |
|---|---|---|---|---|---|---|---|---|---|---|---|---|---|
| 40% | Contra | 68.28 | 23.33 | **10.58** | 2.02 | 78.23 | 31.02 | 8.28 | **0.71** | 80.04 | 15.31 | 6.85 | **1.20** |
| | Fedavg | 68.39 | 14.06 | 10.45 | **5.32** | 80.05 | 31.24 | 10.04 | **9.40** | 79.64 | 35.39 | **5.34** | 12.08 |
| | Foolsgold | 68.52 | 9.80 | 10.25 | **1.75** | 73.45 | 10.32 | 7.89 | **0.87** | 78.31 | 28.20 | 9.84 | **1.00** |
| | Geo med | 54.76 | **20.09** | 23.54 | 30.18 | 78.69 | 41.88 | **22.29** | 54.01 | 79.07 | 73.18 | 76.27 | **64.18** |
| | Krum | 11.08 | **10.28** | 11.38 | 28.80 | 38.07 | 52.79 | **12.25** | 54.64 | 78.33 | 76.77 | **20.98** | 67.66 |
| | **Avg** | 64.99 | 16.82 | **13.71** | 9.82 | 69.70 | 33.45 | **12.15** | 23.93 | 79.08 | 45.77 | **23.86** | 29.22 |
| 20% | Contra | 68.28 | 11.11 | 21.84 | **1.88** | 78.23 | 36.10 | **8.97** | 1.49 | 80.04 | 9.95 | 10.19 | **1.08** |
| | Fedavg | 68.39 | 33.72 | **13.67** | 34.27 | 80.05 | 61.69 | **11.40** | 39.97 | 79.64 | 73.76 | **2.56** | 41.57 |
| | Foolsgold | 68.52 | 25.63 | 10.15 | **3.42** | 73.45 | 42.63 | 10.10 | **1.30** | 78.31 | 11.29 | 4.43 | **1.28** |
| | Geo med | 54.76 | **34.92** | 39.56 | 53.40 | 78.69 | 70.25 | **59.34** | 65.03 | 79.07 | 79.42 | 78.43 | **67.26** |
| | Krum | 11.08 | 18.68 | **10.09** | 26.89 | 38.07 | 66.14 | **17.95** | 54.37 | 78.33 | 78.81 | 78.35 | **67.86** |
| | **Avg** | 64.99 | 26.35 | **21.31** | 23.24 | 69.70 | 55.36 | **21.55** | 32.43 | 79.08 | 50.65 | **34.79** | 35.81 |

[*] We do not include results in Avg when the accuracy of no attack (*NO*) is lower than 30%.

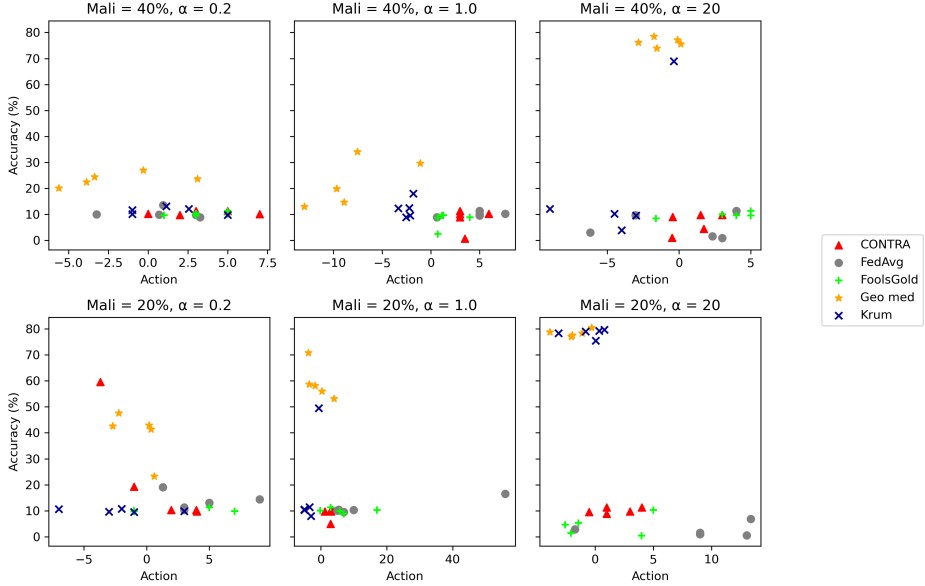

Figure 7: UA-RL action vs accuracy on MNIST

A.4.4 VALIDATION HISTORY ON MNIST WITH ACTION VALUE OF UA-RL

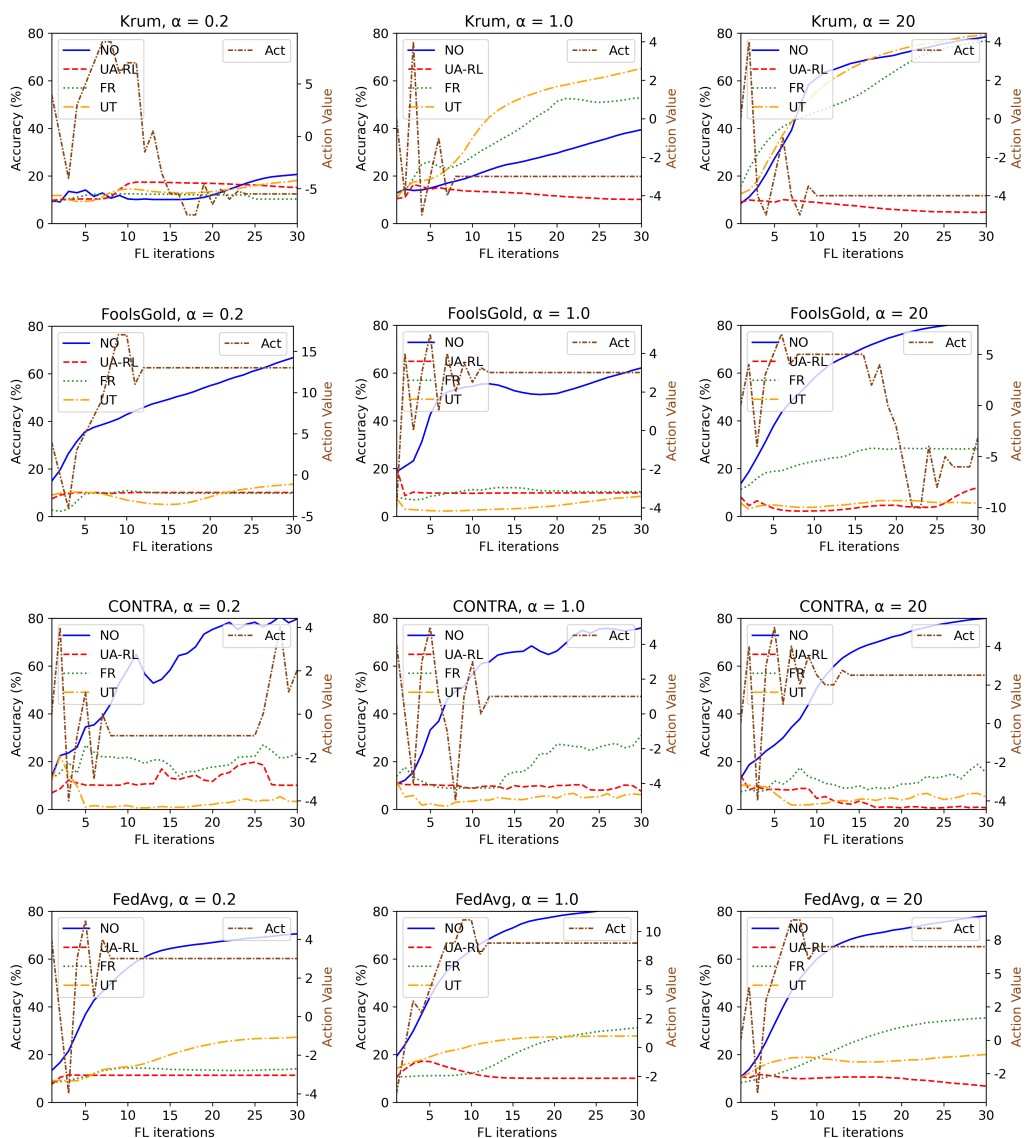

Figure 8: Four aggregation rule's validation history on MNIST with action value of UA-RL. 40% malicious population on Krum, FoolsGold, CONTRA, and FedAvg

