# OpenReview forum: "A Study of Black-Box Attacks Against Robust Federated Learning"
_ICLR.cc/2024/Conference — ICLR 2024 Conference Withdrawn Submission_

### Official Review · Reviewer_4bmv · 2023-10-30

**Soundness:** 2 fair
**Presentation:** 3 good
**Contribution:** 2 fair
**Rating:** 3
**Confidence:** 4

**Summary:**

The paper proposes a new model poisoning attack in federated learning using reinforcement learning to perform untargeted attacks. The authors assumed a black-box model, where the attacker in control of some malicious participants does not have access to the datasets from the rest of the participants and the aggregator rule used by the defender. The authors show that they method is capable of bypassing different robust aggregation algorithms and perform better than some baselines like label flipping attacks or a simplified version of their proposed method without the reinforcement learning component.

**Strengths:**

+ Poisoning attacks in federated learning is a relevant topic. Analyzing attacks trying to use a more realistic model is interesting. For example, the attacker is not assumed to have knowledge about the aggregation algorithm.
+ The theoretical analysis performed at the end of the paper helps to support the benefits of the proposed algorithm.

**Weaknesses:**

The experimental evaluation is not convincing and raises several concerns:
-	The lack of comparison with other attacks proposed in the research literature. The authors argue that none of them have the same threat model, but that is not completely true. For instance, in similar settings as the ones used to perform the experiments, attacks like in Baruch et al. do not really assume knowledge about the aggregation rule or model poisoning attacks as in Bhagoji et al. (“Analyzing Federated Learning through an Adversarial Lens”) could be used in the comparison.
-	The attack is not really strong against standard Federated Averaging, which raises questions about its effectiveness. For instance, Blanchard et al. showed that just with a single adversary, it is possible to completely compromise federated averaging with a trivial attack strategy consisting on sending random model updates with a large variance, whereas, for example, in the experiments reported in Table 1 for MNIST, with 20% of attackers, the accuracy is more than 40% (and up to 70%) for all the results reported for different values of alpha.
-	The results do not include the error bars and the assessment of which method is better in each case should be on the basis of the result of a statistical significance tests.

**Questions:**

+ Compare with other state of the art method like Baruch et al. or Bhagoji et al.
+ Why is the attack not so strong against non-robust aggregation methods like Federated Averaging?
+ Report the error bars and statistical significance tests.

---

### Official Review · Reviewer_cBxt · 2023-10-31

**Soundness:** 3 good
**Presentation:** 2 fair
**Contribution:** 2 fair
**Rating:** 3
**Confidence:** 4

**Summary:**

The paper proposes a black-box model poisoning attack against federated learning (FL). Previous work on model poisoning mainly focuses on white-box and gray-box settings where malicious devices have some knowledge about the FL system, such as the server's aggregation rule and the number of devices and/or the local data of benign agents. The paper adopts a black-box approach where each malicious device independently applies a variant of gradient ascent to identify malicious model parameters to be shared with the server, starting with the global model parameters shared by the server. The main idea of the paper is to search for a scaling parameter for the learning rate using the Nelder-Mead method, which uses the actions and rewards in the last two rounds to search for a new action, where the reward in a round is defined as the decrease of model accuracy in the previous two rounds, where the accuracy is estimated by pooling together the malicious devices' local data. The approach does not require additional information about the server or benign agents.

**Strengths:**

The proposed black-box poisoning attack against FL does not require knowledge about the server's aggregation rule or local data of benign agents and therefore is easier to deploy in practice.

It shows that some commonly used aggregation rules, such as Krum and Geometric Median, are vulnerable to the black-box attack when there are a sufficient amount of malicious devices and local data across devices exhibit a relatively high level of heterogeneity, indicating that these aggregation rules might be insecure to use in certain scenarios.

**Weaknesses:**

The performance of the proposed attack is not very impressive. Although it typically outperforms the two simple baselines, they are comparable with small gaps in most cases. Further, with 20% malicious devices, neither of them can compromise the system using vanilla FedAvg when the local data distributions are close to iid, and neither of them can compromise the geometric median unless the local data distributions are highly non-iid. Even with 40% malicious devices, which is far from being realistic, they cannot compromise the geometric median when the local data distributions are close to iid. Why should one still worry about the black-box attacks?

The evaluation is not comprehensive. First, the two commonly used robust aggregation rules, namely, trimmed mean and coordinate-wise median, are not considered. Second, the paper assumes that all the devices are sampled in each round. However, subsampling is typically used in large FL systems. Since the Nelder-Mead method relies on actions and rewards from consecutive rounds, it is unclear if the proposed attack is still effective with subsampling. Third, the paper does not consider more advanced defenses such as FLTrust (Cao et al., 2021), which requires the server to access a small amount of clean data but can potentially provide better protection to the FL system.

Xiaoyu Cao, Minghong Fang, Jia Liu, Neil Zhenqiang Gong. FLTrust: Byzantine-robust Federated Learning via Trust Bootstrapping. NDSS 2021.

**Questions:**

The paper does not define the attacker's Markov decision process and the attack decision in each FL round seems myopic. Why is the attack called an RL-based attack?

Can the Nelder-Mead-based solution provide any performance guarantee to the attacker's problem?

In Algorithm 1, how is the initial set of actions in E defined?

---

### Official Review · Reviewer_hDgi · 2023-10-31

**Soundness:** 1 poor
**Presentation:** 2 fair
**Contribution:** 2 fair
**Rating:** 3
**Confidence:** 3

**Summary:**

This paper introduces a new black-box attack for Byzantine adversaries in Federated Learning. The introduced attack makes use of reinforcement learning methods to bypass several defenses. Empirical results are provided in iid and non-iid settings to showcase the strength of the proposed attacks. Theoretical arguments are also given to explain the empirical performances.

**Strengths:**

The development and analysis of new attacks in distributed machine learning is a valuable contribution to the community. This is even more the case if it is done with minimal knowledge, as in the black-box setting considered by the authors.

**Weaknesses:**

There are critical weaknesses suggesting that the paper is not yet ready for publication, mainly concerning the inaccuracy of the main claim of the paper, the great lack of clarity, and the weakness of the experimental validation.

### 1. The proposed attack is not black-box as claimed.

In Section 3, it is explained by the authors that action taken by the adversary depends on the previous rewards. Computing these rewards requires being able to evaluate the accuracy of the server's model on the test set. To do so, the adversary must have access to the data of honest workers, which means that the attack is not black-box, even according to the paper's problem setup in Section 3.1. Also, the attack implicitly assumes the knowledge of the optimizer used by the server; a constant learning rate is used throughout the paper.

Given that this is a central claim of the paper, this is my main concern.

### 2. The paper is extremely unclear in several instances.

* The reinforcement learning branding seems unneeded, as far as I understand, and only adds clarity-affecting complexity. It seems that the proposed attack tries to solve an optimization problem to find the attack vector at each step, which should be explained as such. The formalism of action and rewards is unnatural here, especially given that the adversary does not learn a policy (mapping states to actions) but rather actions only when needed.

* The last paragraph of Section 3 mentions some specific behavior of the algorithm in the first $d$ (undefined before use) steps, but does not explain/justify this algorithm design, which currently seems to be arbitrary.

* Several notation issues: in Section 3.2 the summations are indexed by $i$, but the summands are indexed by $j$ and not $i$, in Section 5.1, $s(i)$ is defined for all $i$ but not $s_o(i)$ and they seem to stand for the same quantity.

* I found Theorem 1 hard to parse, and wonder what are the takeaways from it. Specifically, the use of mathematical quantifiers is strange (and $\exists$ is denoted as $\ni$), and I do not really know how to interpret the result. For example, (3) is an inequality lower bounding a non-negative quantity by a quantity that could be negative (if some of the cosines are negative).

### 3. The experiments are insufficient.

* The defenses used are not designed for the non-iid setting, which makes the comparison ill-posed. There are methods (e.g., NNM [1], Bucketing [2]) specifically designed for this purpose and should be included to enable a full experimental validation.

* Other black-box (or grey-box) attacks should be included for completeness, e.g., the attacks of Baruch et al. (2019) and Li et al. (2022).

* Minor comment: the choices of the Dirichlet parameter $\alpha$ are a little strange, it is typical to do $0.1, 1, 10$.

References:

[1] Allouah et al., Fixing by Mixing: A Recipe for Optimal Byzantine ML under Heterogeneity, AISTATS 2023.

[2] Karimireddy et al., Byzantine-Robust Learning on Heterogeneous Datasets via Bucketing, ICLR 2022.

**Questions:**

Please address the weaknesses above.

---

### Official Review · Reviewer_UYNs · 2023-11-04

**Soundness:** 2 fair
**Presentation:** 2 fair
**Contribution:** 2 fair
**Rating:** 3
**Confidence:** 4

**Summary:**

In this submission, the authors propose an untargeted attack algorithm, called UA-RL, to enhance the robustness of federated learning in a black-box setting. The proposed UA-RL adopts a gradient ascent technique to maximize the loss function, and utilizes reinforcement learning algorithms to optimize the magnitude of the malicious gradients.

**Strengths:**

1. The authors study an interesting and practical problem in this paper.
2. The proposed method UA-RL is described clearly, and the authors will release the source code to further promote the research in the community.

**Weaknesses:**

1. The writing of this paper should be further polished. (a) Typos: MINST-> MNIST; (b) The relationship between the ``Discussion and Analysis'' section and other sections (especially the proposed method) in this submission is not clear.
2. Some important references are missed in the submission, such as [1][2]. The settings and proposed method in [1] are closely related to UA-RL, and the authors can provide more discussions and highlight the advances of UA-RL.
3. The experiments are conducted on three similar datasets, MNIST, Fashion-MNIST, EMNIST. More diverse datasets can be considered to make the experiments more convincing.
4. Some experimental results in Table 1 are confusing. For example, (a) Why Krum (w/o attack) performs much worse than other baselines, and even worse than Krum (w/ attack).
? e.g., 10%+ v.s. 50%+ when $\alpha=0.2$; 40% v.s. 70%+ when $\alpha=1.0$. (b) Some naive baselines, such as fixed reverse gradient (FR) and untargeted label flipping (UT), seem to perform better than UA-RL in 11/30 cases in Table 1.

References:
[1] Manipulating the Byzantine: Optimizing Model Poisoning Attacks and Defenses for Federated Learning. In NDSS, 2021.
[2] Learning to Attack Federated Learning: A Model-based Reinforcement Learning Attack Framework. In NeurIPS, 2022.

**Questions:**

Please refer to the Weaknesses above.